# The Methodological Quality of Studies on Physical Exercise in Adolescents with Cerebral Palsy: A Scoping Review of Systematic Reviews and Meta-Analyses

**DOI:** 10.3390/healthcare12202039

**Published:** 2024-10-15

**Authors:** Alexandrina Cavalcante Rodrigues Nitz, Maria João Campos, Ana Amélia Moraes Antunes, Emilly da Silva Freitas, Chrystiane Vasconcelos Andrade Toscano, José Pedro Ferreira

**Affiliations:** 1Faculty of Sport Sciences and Physical Education, University of Coimbra, 3040-248 Coimbra, Portugal; 2Sarah Network of Hospitals of Rehabilitation, Fortaleza 60861-634, CE, Brazil; 3Research Unit for Sport and Physical Activity (CIDAF, uid/dtp/04213/2020), Faculty of Sport Sciences and Physical Education, University of Coimbra, 3040-248 Coimbra, Portugal; mjcampos@fcdef.uc.pt (M.J.C.); jpferreira@fcdef.uc.pt (J.P.F.); 4Sarah Network of Hospitals of Rehabilitation, Belo Horizonte 30510-000, MG, Brazil; antunesanaamelia@gmail.com; 5Physiotherapy Department, Federal University of Ceará, Fortaleza 60430-160, CE, Brazil; emilly.freitas00@hotmail.com; 6Research Project on Physical Exercise for People with Autism Spectrum Disorder (PEFaut), Institute of Physical Education and Sport, Federal University of Alagoas (UFAL), Maceió 57072-970, Al, Brazil; chrystoscano@gmail.com

**Keywords:** physical activity, physical fitness, well-being, neurodevelopment disorder, teenager, rehabilitation

## Abstract

Introduction: Systematic reviews and meta-analyses point to the benefits of physical exercise for adolescents with cerebral palsy, improving physical conditioning, muscle strength, balance, and walking speed. However, given the high number of reviews that include randomized and non-randomized studies, it is increasingly necessary to assess the methodological quality of these reviews. This scoping review investigated the methodological quality of systematic reviews and meta-analyses on the effects of physical exercise in adolescents with cerebral palsy to elucidate the methodological limitations of the research and the priorities to be observed in future research. Method: The electronic search used PubMed, Web of Science, and Cochrane. Studies published between 2016 and 2023 were selected. The terms used were “cerebral palsy” combined with “physical fitness”, “exercise”, and “physical activity”. Results: A total of 219 original reviews were selected. Of these, 19 reviews were included for data analysis. AMSTAR2 was used to assess the methodological quality of the reviews. Three reviews presented high methodological quality (15.78%) and three had moderate methodological quality (15.78%). The remaining reviews had low or critically low methodological quality, according to AMSTAR2. Interpretation: This study evidenced that systematic reviews have variable methodological quality and that new studies are still needed.

## 1. Introduction

Physical exercise (PE) is defined as any planned, structured, repetitive, and completed activity intended to improve physical fitness [1]. Over the years, other benefits of PE have become clear and valued by the general population, regardless of personal and environmental factors [2,3,4].

There is evidence that aerobic exercise contributes to improving known risk factors for cardiovascular diseases [5]. In adolescents, in addition to the physical benefits, it is also accepted that PE can have positive effects on mood, stress, and anxiety [6,7]; is effective in the prevention and treatment of depression [8]; and is associated with reduced symptoms of mental disorders [9].

Exercise has also been an integral part of treating and rehabilitating many medical conditions [10], especially those whose therapeutic objectives include improving motor function, such as in cerebral palsy (CP) [11]. CP is a group of permanent but not immutable disorders of movement and/or posture and motor function due to a non-progressive interference, injury, or abnormality of the developing/immature brain [12]. Some research shows that individuals with CP who exercise and play sports experience improved health benefits, such as cardiorespiratory endurance [13,14,15]. Studies also point to the benefits of some types of muscle-strengthening exercises in adolescents with CP, such as strength gain and improvement in motor activity and walking ability [16,17]. Functional training with free weights, for example, has been shown to be safe and effective in increasing physical performance parameters and cardiovascular health in adolescents with high-functioning CP [18]. Despite this, during the transition to adulthood, young people with CP experience decreasing physical activity levels [19]. In addition, CP is a complex and comprehensive diagnosis [20]. The heterogeneity of cognitive, behavioral, and motor characteristics results in highly variable studies [20]. The small sample sizes and the failure or loss of participants could reduce the potential effect size of an intervention [14,20]. Challenges to identifying effective treatments also include a lack of participants with more significant disabilities and the use of outcome measures that are not uniformly adopted or used as intended [14,20].

Systematic reviews and meta-analyses attempt to illustrate the potential role of exercise interventions for children and adolescents with CP, which are widely used for clinical decisions. However, there has been an increase in the number of non-randomized studies included in reviews [21]. In this way, this study evaluated the methodological quality of systematic reviews and meta-analyses of the effects of PE in adolescents with CP to elucidate the methodological limitations of the research, seek continuous improvement of the review processes on the topic, and identify priorities to be observed in future research. This study also synthesized the results of systematic reviews, which were classified as having high and moderate methodological quality, and assessed the level of evidence of these reviews to help direct rehabilitation practices for this population.

## 2. Materials and Methods

### 2.1. Protocol and Guidelines

This scoping review followed the PRISMA-ScR guidelines [22]. The protocol for this review was registered in an international registered PROSPERO database, registration number CRD42023465691.

### 2.2. Search Strategy

The search strategies were based on the following descriptor terms and keywords defined by the authors and indexed to the Medical Subject Headings (MESH, U.S. National Library of Medicine, 8600 Rockville Pike, Bethesda, MD, USA): “cerebral palsy” combined with “physical fitness” AND exercise AND “physical activity”. Combinations of these keywords were entered into searches in the following academic journal databases: U.S. National Library of Medicine National Institutes of Health (U.S. National Library of Medicine, Rockville Pike, Bethesda, MD, USA; https://www.ncbi.nlm.nih.gov/pubmed/, with last access in 30 December 2023); Web of Science (Clarivate Analytics, Boston, MA, USA; http://apps.webofknowledge.com, with last access in 30 December 2023); and Cochrane Database of Systematic Review (https://www.cochranelibrary.com/search, with last access in 30 December 2023). The advanced metasearch option was carried out using the resources inherent to each database. The search included systematic reviews and meta-analyses published in English between 1 January 2016 and 1 December 2023.

The research procedures were conducted between 1 September 2023 and 30 December 2023.

### 2.3. Selection Criteria

The inclusion criteria for this scoping review were based on the “PICOS” criteria:Patients: Systematic reviews or meta-analyses in which at least 50% of the included articles had a sample composed of adolescents (10–19 years old).Intervention: Physical exercise. In this study, physical exercise was defined as any activity that is planned, structured, repetitive, and purposefully completed to improve physical fitness [1].Comparison: Comparison before and after the intervention or comparison with any other intervention, as long as the target population received intervention with some physical exercise program.Results: Physical fitness. For this study, physical fitness was considered a set of attributes related to health or skills, with the health-related components of physical fitness being cardiorespiratory endurance, muscular endurance, muscular strength, body composition, and flexibility [1]. Programs with physical exercise, which had other domains as outcome measures, in addition to physical fitness, were also considered, as long as physical fitness was one of the outcome measures.Of-Study Design: Systematic review or meta-analysis only.

Exclusion criteria included preprints, unpublished data, narrative or literature reviews, and papers for which the authors of this review could not access the full text.

### 2.4. Data Collection

A three-step screening process was implemented to determine whether initial search articles were relevant.

Step 1: Two authors independently performed the initial searches by dividing the search engines and using the list of keywords for the analysis. Relevant titles were saved (Microsoft Excel 365) and transferred to a citation program (Endnote X9). After completion of the initial screening, all duplicate titles were removed.

Step 2: Two authors independently screened the abstracts of the selected articles, and if the abstract did not meet the inclusion criteria, it was excluded from the study.

Step 3: Two authors independently retrieved the remaining full-text articles for further screening. If the articles did not meet the inclusion criteria or did not provide sufficient information related to the inclusion criteria, they were excluded from the study. Disagreements between authors during the second or third stage were discussed with a third author until a consensus was reached.

For all studies, information of interest included publication characteristics (e.g., author, year of publication), participant characteristics (e.g., sample size, age, diagnoses), and main results. All information was collected individually by two reviewers. A third reviewer cross-checked the information.

### 2.5. Methodological Quality Assessment

Two independent reviewers assessed the methodological quality of each study using AMSTAR2 [21]. Any inconsistencies between the two reviewers were resolved by discussion or a third person if disagreements persisted.

AMSTAR2 is a tool developed to evaluate randomized and non-randomized intervention study reviews. It aims to assist decision-makers in identifying high-quality systematic reviews. It contains 16 items in the form of questions that can be answered with yes, partial yes, no, or no measure. The tool classifies items as critical or non-critical.

The seven critical issues are as follows: Protocol registered before the commencement of the review (item 2); Adequacy of the literature search (item 4); Justification for excluding individual studies (item 7); Risk of bias from individual studies being included in the review (item 9); Appropriateness of meta-analytical methods (item 11); Consideration of risk of bias when interpreting the results of the review (item 13); Assessment of presence and likely impact of publication bias (item 15). The others are considered non-critical.

The overall confidence of a review can be categorized as high if the systematic review does not present any or presents a single “NO” answer for an item considered non-critical; moderate if the systematic review presents more than one “NO” answer on items considered non-critical; low if the systematic review presents at least one “NO” answer on a critical item, with or without a “NO” answer for items considered non-critical; and critically low if the systematic review presents more than one “NO” answer for critical items, with or without “NO” answers for non-critical items. The 16 items of AMSTAR2 are presented in the Appendix A.

### 2.6. Summary and Quality of Evidence (GRADE)

After carrying out AMSTAR2, the reviews were summarized and classified as having high and moderate methodological quality. The overall certainty of the evidence of these reviews (high and moderate methodological quality) was evaluated using the Grading of Recommendations Assessment, Development, and Evaluation (GRADE) system [23]. When available, the GRADE made by the authors was used. The GRADE criteria were assessed by consensus of two independent reviewers. A third person resolved any inconsistencies between the two reviewers if disagreements persisted.

The GRADE system assesses the risk of bias, inconsistency of results, indirectness of evidence, imprecision, and risk of publication in individual studies and classifies the certainty of evidence into four levels: High (the findings are robust), Moderate (new research may change our results), Low (the level of confidence in our pooled effect is very slight), and Very Low (any estimate of effect is very uncertain).

## 3. Results

The search yielded 219 reviews: Web of Science, 138; PubMed, 74; Cochrane, 5; and 2 reviews were added based on references from the original reviews found. Although the language was limited to English, an article in Spanish was added to the scope through citation searching. Duplicate titles (51) were removed. Scoping/analytical reviews and studies with other interventions, diagnoses, and populations or no outcome measures were also removed (147). Twenty-one reviews were assessed for eligibility, but two of them were excluded because the authors of this review could not access the full text. Nineteen reviews were included in this study. Figure 1 shows a flow diagram of review selection.

### 3.1. Characteristics of the Studies

Nineteen reviews were analyzed, five of which were systematic reviews and fourteen were meta-analyses. The interventions with PE were varied, as were the assessments and expected outcomes. One review investigated the effects of cycling [24], one investigated the effects of dancing [25], two investigated the effects of aquatic exercise [26,27], and four investigated exercise with virtual reality [28,29,30,31]. The remaining reviews investigated the effects of general exercises, such as resistance and strengthening [32,33,34,35,36,37,38,39,40,41,42].

Concerning the four virtual reality reviews, only one used a balancing platform associated with the virtual resource balance board on the Nintendo Wii [28]. The others included systems that utilize the concept of virtual reality regardless of the degree of immersive experience (e.g., Sony PlayStation), input device (e.g., Kinect, motion camera), or display device (e.g., TV screen, curved projection screen).

The primary outcomes were gains in muscle strength, walking speed, gross motor function, aerobic capacity, balance, and participation.

A complete description of the articles is found in Table 1.

### 3.2. Evaluation of Methodological Quality

#### 3.2.1. Analysis of Negative Responses

In a global analysis of the selected evaluations, the items for which the evaluations presented the highest number of negative responses were 10, 12, 13, 14, and 15.

The funding sources for the studies included in this review were not reported in seventeen reviews [24,25,26,27,29,31,32,33,34,35,36,37,38,39,40,41,42] (item 10 of AMSTAR 2).

The authors did not evaluate the potential impact of risk of bias in individual studies on the meta-analysis results or other evidence synthesis in nine reviews [24,25,28,29,31,35,40,41,42] (item 12 of AMSTAR 2).

The authors did not rely on the risk of bias to interpret/discuss the review results in ten reviews [24,25,26,27,31,33,35,39,40,41,42] (item 13 of AMSTAR 2).

The authors did not provide a satisfactory explanation or discussion of any heterogeneity observed in the review results in eight reviews [25,26,27,33,39,40,41,42] (item 14 of AMSTAR 2).

They also did not perform a quantitative synthesis adequately investigating publication bias or discussing its likely impact on the review results in ten reviews [24,25,29,32,33,35,39,40,41,42] (item 15 of AMSTAR 2).

#### 3.2.2. Analysis of Critical Items

In a specific analysis of the critical items of AMSTAR 2, it was observed that the review methods were established before the review was conducted, and the report justified any significant deviations from the protocol so that the reviews did not present negative responses to item 2 of AMSTAR 2.

The authors of all reviews used a comprehensive literature search strategy, achieving at least a “partially yes” response to item 4 of AMSTAR 2.

In four reviews, the authors did not provide a list of excluded studies or did not justify their exclusions [30,33,41,42] (item 7 of AMSTAR 2).

In three reviews, the authors did not use satisfactory techniques to assess bias risk in the selected individual studies [27,32,35] (item 9 of AMSTAR 2).

Five meta-analyses did not present appropriate methods for a statistical combination of results [25,39,40,41,42] (item 11 of AMSTAR 2).

Items 13 and 15 were addressed in the global analysis.

It is important to note that systematic reviews without meta-analyses are not evaluated by items 11, 12, and 15. This exception is based on the understanding that these items specifically pertain to the process and results of meta-analyses and therefore do not apply to reviews without this component.

Of the nineteen reviews, three (15.78%) had high methodological quality and three (15.78%) had moderate methodological quality. The remaining papers had low or critically low methodological quality, according to AMSTAR 2. Table 2 shows the AMSTAR 2 for each article and the results.

### 3.3. Description of Reviews of High Methodological Quality

#### 3.3.1. Intervention: Power Exercise

A recent review by Özgün et al. [34] investigated the effectiveness of power exercises on muscle strength, walking speed, gross motor function, and activities of daily living. Of the ten studies selected, three were compared to a traditional strength training intervention, six were compared to routine physical therapy, and one was compared to a resting condition. When comparing power exercise with traditional strength training, it was impossible to conclude which one was better at increasing power, length, and muscle strength since the results reported by the articles were conflicting. Compared to traditional physiotherapy, conflicting evidence was found regarding muscular power, walking resistance, and mobility; moderate evidence was found for more significant improvement in muscular strength; and weak evidence was found for improving school participation. Compared to resting condition, there are no statistics for the effect of power exercise. Limited evidence of pain reduction favoring power exercise has been demonstrated. In conclusion, there is moderate evidence showing that power training improves muscle strength, walking speed, gross motor function, and activities of daily living. The authors suggested that the lack of more substantial evidence for power exercise interventions improving muscular architecture, muscle function, walking ability, and mobility in children with CP may be explained by differences in training protocols and the degree to which these met the physiological definition of power, the different methods of measuring power, limited durations of training, and the relative effectiveness of control interventions.

Quality of evidence (GRADE): MODERATERisk of bias: moderateInconsistency: no informationIndirectness: highImprecision: moderatePublication bias: low

#### 3.3.2. Intervention: Aerobic Exercise

A systematic review by Czencz et al. [38] investigated the effects of exercise on one or more components of health-related physical fitness, including cardiorespiratory endurance, muscular endurance, strength, body composition, and flexibility. The outcomes of interest were quality of life and participation. The results indicated that aerobic exercise improves aerobic capacity, balance, gross motor function, mobility, and participation. However, aerobic exercise is less effective than usual care or other interventions in improving muscle strength, spasticity, gait parameters, and quality of life.

Quality of evidence (GRADE): MODERATE-LOWRisk of bias: lowInconsistency: lowIndirectness: highImprecision: moderatePublication bias: moderate

#### 3.3.3. Intervention: Resistive Therapy

A meta-analysis by Collado-Garrido et al. [36] analyzed the impact of resistive therapy on improving gait. Ten articles were selected, nine randomized controlled clinical trials and one uncontrolled clinical trial. All ten studies provided data relating to a resistive therapy intervention that is amenable to meta-analysis for assessing pre-post intra-group differences in gait speed, and six of them also provided data relating to stride frequency and length. The overall effect was favorable to the intervention. The results support the impact of resistive therapy on improving gait, especially in terms of gait speed and step cadence parameters, with a small effect size. The study shows that resistance therapy would increase muscle strength in children with CP. This increase in strength would have an impact on improving gait.

Quality of evidence (GRADE): MODERATERisk of bias: moderateInconsistency: moderateIndirectness: highImprecision: moderate to highPublication bias: low

### 3.4. Description of Reviews of Moderate Methodological Quality

#### 3.4.1. Intervention: Nintendo Wii Therapy (NWT) with Balance Board

A meta-analysis by Montoro-Cardenas et al. [28] analyzed the efficacy of NWT on functional balance in children with CP. They included 11 studies with reported data of 270 children and adolescents. They reported that NWT represents an effective treatment for functional and dynamic balance in children with CP. Evidence of moderate quality was found for a significant effect of NWT on functional balance. Subgroup analyses revealed moderate-quality evidence for a medium effect of using NWT plus conventional physiotherapy versus conventional physiotherapy alone in functional balance.

Quality of evidence (GRADE): LOWRisk of bias: moderateInconsistency: lowIndirectness: lowImprecision: lowPublication bias: moderate

#### 3.4.2. Intervention: VRT (Virtual Reality Therapy)

A meta-analysis of randomized controlled trials by Wu, Loprinzi, and Ren [31] evaluated the effect of VRT games on balance recovery in children with cerebral palsy. They looked at 11 studies, each involving 16 to 48 teenagers. Randomized controlled trials were selected and underwent meta-analysis. The review reported that VRT games played a positive role in improving the balance of children with CP. Still, these results should be viewed cautiously due to current methodological defects (measurement difference, heterogeneity of control groups, intervention combined with other treatments).

Quality of evidence (GRADE): MODERATE TO HIGHRisk of bias: moderateInconsistency: moderateIndirectness: highImprecision: highPublication bias: moderate

#### 3.4.3. Intervention: Aerobic Exercise

A meta-analysis by Soares, Gusmão, and Souto [37] investigated the efficacy of aerobic exercise on the body functions, structures, activities, participation, and quality of life of children and adolescents with CP. They included fifteen randomized controlled trials with 414 participants with CP. The effect of aerobic exercise compared to usual care or other interventions was significant for aerobic capacity, mobility, balance, and participation. Aerobic exercise was ineffective for muscle strength, spasticity, gait parameters, and quality of life.

Quality of evidence (GRADE): MODERATE TO LOWRisk of bias: highInconsistency: moderateIndirectness: highImprecision: lowPublication bias: low

## 4. Discussion

According to the findings of previously mentioned papers, PE promotes several physical benefits [14,15,16,17]. However, studies are highly variable and many gaps still need to be filled [20].

This scoping review describes the methodological quality of systematic reviews and meta-analyses of articles on the effects of PE in adolescents with CP to elucidate the methodological limitations of research, seek continuous improvement of the review processes on the topic, and identify priorities to be observed in future research.

### 4.1. Main Findings

The systematic reviews and meta-analyses analyzed were diverse regarding the exercises studied, number of studies included, type of studies included (randomized/non-randomized), and number of participants in each study.

The number of studies included in the review, whether the review included only randomized studies, and the number of participants in the studies included in each review were not factors that interfered with the result in the assessment of methodological quality.

The systematic reviews and meta-analyses evaluated were diverse in terms of exercise protocols. Thus, it was impossible to identify whether review articles on a specific exercise modality presented better methodological quality than those on another.

Of the 19 reviews selected, three had high methodological quality (15.78%) and the other three had moderate methodological quality (15.78%), according to AMSTAR2. Approximately 68% of the reviews presented low or critically low methodological quality.

Analyzing only the critical items of AMSTAR2, the main methodological problems identified were risk of bias and heterogeneity.

Some reviews could have used a more satisfactory technique to assess the risk of bias (item 9 of AMSTAR2), such as genuinely randomized allocation, outcome selection among multiple measures, or analysis of a given outcome. Some authors should have explained the likely impact of the risk of bias of individual studies when interpreting/discussing the review results (item 13 of AMSTAR2) and pointed out whether they would be severe enough to lower the certainty of the evidence.

The absence of a quantitative synthesis was a significant gap in a few meta-analyses. The authors did not adequately investigate publication bias or discuss its likely impact on the review’s results (item 15 of AMSTAR2). The presentation of a funnel plot or statistical test of publication bias (e.g., Egger or Beggs tests) and a discussion of the probability and magnitude of the impact of publication bias are crucial steps that should not be overlooked.

Regarding non-critical items, some authors, although they analyzed the risk of bias, did not provide an analysis of the impact of these risks on the results or estimates of combined effects, such as sensitivity analysis (item 12 of AMSTAR2). As for heterogeneity, when investigated, some studies did not associate it with the investigation of possible sources, nor did they discuss the impacts on the results (item 14 of AMSTAR2). Item 8 of the instrument focuses on adequately describing the instruments under analysis, and it is concerning that few reviews reported a complete follow-up. Given that the objective of some studies was to evaluate strength gain and functional parameters, the evaluation of these parameters was not just relevant but crucial. This should be a key area of focus for researchers. Equally important is reporting the sources of funding for the included studies, a practice that only two reviews adhered to (item 10 of AMSTAR2).

Most reviews with high and moderate methodological quality had a moderate quality of evidence (ranging from moderate-low to moderate-high) to demonstrate the intended effects. The certainty of evidence refers to how certain an intervention’s effect is within a chosen range. Considering that we only evaluated reviews with the best methodological quality, that CP is a complex and comprehensive diagnosis, and the challenges of implementing PE programs in adolescents with this diagnosis, we believe that this result is favorable to PE intervention.

### 4.2. Other Considerations

Other reviews that used AMSTAR2 as an instrument to evaluate the methodological quality of intervention with exercise also found mixed results [43,44,45,46]. Reviews pointed out that many systematic reviews and meta-analyses did not explicitly state that the review methods were established a priori or provide justification for excluded articles or identification of a funding source.

The varied intervention protocols in the evaluated systematic reviews and meta-analyses may also have contributed to the results. Other reviews also reported finding diverse recruitment strategies, assessment parameters, outcome measures, and intervention options for each exercise modality, such as varying frequency and intensity in the reviews analyzed [43,44,45,46].

We suggest that future randomized clinical trials preferably include larger cohorts, well-defined control groups, and replicable and generalizable evaluation and intervention parameters. Systematic reviews should follow the PRISMA or Cochrane guidelines and analyze and report the level of evidence so the reader can value the results. 

We also suggest that decision-makers in clinical practice use AMSTAR2 to assess systematic reviews and meta-analyses.

This review has several limitations. Although this study adhered to the criteria and recommendations for conducting a scoping review, this review excluded systematic reviews and meta-analyses with some types of physical exercise because they did not meet the inclusion criteria (such as language or year of publication). Considering the large number of types of physical exercise, the authors did not search for isolated terms to avoid excluding any exercise due to forgetfulness or lack of knowledge. Therefore, this review did not include some types of exercises because they were not found with the search terms used.

We strongly suggest that our findings be interpreted within the context of the limitations of our study.

## 5. Conclusions

The methodological quality of systematic reviews and meta-analyses on the effects of physical exercise in adolescents with CP is variable, but only 15.78% have high methodological quality.

New studies are still needed.

As far as we know, this study is the only one that has evaluated the methodological quality of systematic review and meta-analysis studies on physical exercise specifically in adolescents with cerebral palsy using the AMSTAR2 instrument in the last few years.

What this study adds:-The systematic reviews and meta-analyses on the effects of physical exercise in adolescents with CP were diverse regarding the exercises studied, the number of studies included, the type of studies included (randomized/non-randomized), and the number of participants in each study.-The number of studies included in each review, whether the review included only randomized studies, and the number of participants included in each review were not factors that interfered with the result in assessing methodological quality.-The methodological quality of systematic reviews and meta-analyses on the effects of physical exercise in adolescents with CP is variable, but only 15.78% have high methodological quality.

## Figures and Tables

**Figure 1 healthcare-12-02039-f001:**
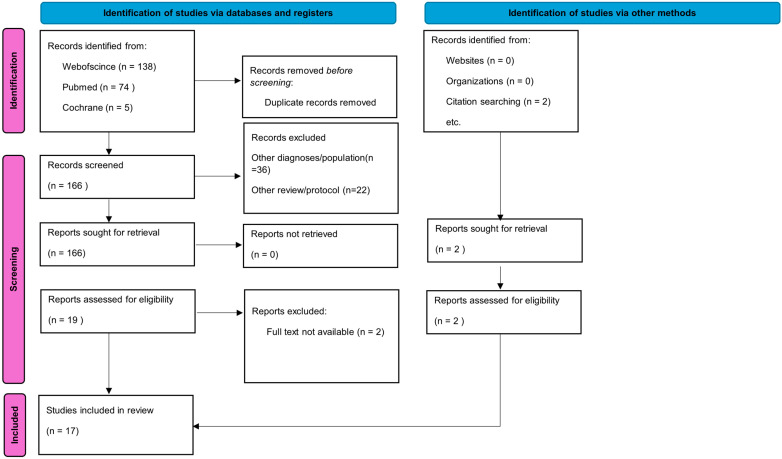
Flowchart.

**Table 1 healthcare-12-02039-t001:** Study characteristics.

Reference Number	Database	Studies	Population	Exercise	Outcome	Measure	Conclusion
[24]	SCOPUS, PubMed, CINAHL, EMBASE, SPORTDiscus, and Cochrane.	Nine articles in qualitative and four in quantitative synthesis; 1 to 62 participants per study.	Cerebral palsy, 6–14 years.	Cycling	To improve muscle strength, balance, and gross motor function in children with cerebral palsy.	GMFM; PEDALS; PODCI; PEDI-CAT.	Significant improvements were reported for hamstring strength (effect size = 0.77–0.93), cardiorespiratory fitness (effect size = 1.13–1.77), balance (effect size = 1.03–1.29), 3 min walk test distance (effect size = 1.14), and gross motor function (effect size = 0.91). Conclusions were limited by small sample sizes, inconsistent outcome measures, and lack of follow-up testing.
[42]	EMBASE, PubMed, Scielo, PEDro, and CINAHL.	Ten articles (RCTs); 19 to 68 participants per study.	Cerebral palsy, 5–24 years.	Aerobic training, progressive resistive exercises, and treadmill training.	Improve participation in exercise, sport, and physical activity.	PEM-CY; Life-H; Participation in physical activities: CAPE; COPM.	Positive pooled effects on leisure time participation were seen only in group interventions at short-term follow-up compared with control. Sports-focused interventions did not improve overall participation.
[34]	MEDLINE (PubMed), Cochrane, and Web of Science.	Ten articles; 10 to 42 participants per study.	Cerebral palsy, 4–20 years.	Power training.	Walking speed, activities of daily living, muscle strength, and gross motor function.	1MFWT; 6MWT; TUG; GMFM; PODC; ASK; MMT; FMS; Goal Attainment Scale; 3D Gait analysis; PEM-CY; dynamometer.	Moderate evidence was found that strength exercises increased walking speed, activities of daily living, and muscle strength and improved gross motor function more than a routine physical therapy program. The lack of stronger evidence might be explained by the differences in training protocols and the degree to which these met the physiological definition of power, the different methods of measuring power, limited durations of training, and the relative effectiveness of control interventions.
[35]	PubMed, Embase, CINAHL, and Web of Science.	Forty-one studies (11 RCTs); 1 to 95 participants per study.	Cerebral palsy, 5–25 years.	Functional gait training.	Walking speed was the most commonly reported gait outcome.	2MFWT; 6MWT; 10MWT; GMFM, TUG; PEDI.	There is promising evidence that functional gait training is a safe, feasible, and effective intervention to target improved walking ability in children and young adults with CP. The addition of virtual reality and biofeedback can increase patient engagement and magnify effects.
[36]	Medline, ISI Web of Knowledge, and PEDro.	Ten articles (nine controlled studies and one single-arm study); 6 to 62 participants per study.	Cerebral palsy, 4–18 years.	Resistive therapy.	Impact on strength, motor function, and gait (at least one gait parameter).	GMFM; STS; LSU; MobQue; TUG.	The results concluded that resistive therapy improves muscle strength and gait parameters, and the study established an intervention protocol of 40–50 min for three days a week.
[41]	PubMed, Embase, and Cochrane.	Twenty-seven studies; 12 to 101 participants per study.	Cerebral palsy, 6–16 years.	Aerobic or resistance training.	Gross motor function, gait speed, and muscular strength.	ICF; GMFM; Three-dimensional gait analysis; Biodex dynamometer; The Biodex Gait Trainer 2TM; WeeFIM; Bruininks– Oseretsity test; Lateral step up Test; 6MWT; GAS.	The meta-analysis revealed that exercise interventions are not associated with improved gross motor function in children with CP, but were associated with increased gait speed and muscle strength.
[39]	PubMed, Cochrane, Science Direct, OVID, ERIC, Dare, and PEDro.	Nine studies (RCTs); 8 to 30 participants per study.	Cerebral palsy, 10–12 years.	Activity training on the ground (such as sit-to-stand, walking, or stepping).	Most of the studies used either the Gross Motor Function Measure or the Timed Up and Go.	1MFWT; GMFM; TUG; MAS; Wee-FIM.	The available evidence shows little effect of activity training on the ground on activity or participation in children with CP, suggesting that rigorous trials with larger samples and a larger “dosage” of activity training on the ground are needed in the future.
[38]	Cochrane, MEDLINE, Embase, CINAHL, Pedro, Science Web, Otseeker, SCOPUS, and PubMed.	Seventeen studies; 6 to 95 participants per study.	Cerebral palsy, age ≥ 16 years.	Strength, aerobic, or treadmill training; dance; and swimming.	Quality of life, participation, pain, mood, and fatigue.	LIFE-H; SF-36; WHOQOF-BREF; 10WMT, FIM; GMFM.	The review could not answer the primary questions about the effect of an exercise intervention on quality of life and participation nor the secondary questions about pain, mood, fatigue, and self-efficacy. A limited number of studies suggest that exercise may improve fatigue and mental health in adults with CP, but more research is needed to confirm these findings.
[37]	Embase, PubMed, CINAHL, and PEDro.	Fifteen RCTs; 6 to 34 participants per study.	Cerebral palsy, 4–21 years.	Aerobic exercise.	Aerobic capacity, muscle strength, spasticity, balance, gait parameters, gross motor function, mobility, quality of life, and participation.	6MWT; 10MWT; Shuttle Run Tests; Muscle Power Sprint Test; computerized dynamometer; Second Sit to Stand Test; MAS; BBS; Biodex Gait; three-dimensional motion analysis; GMFM.	Aerobic exercise improves gross motor function but not gait parameters in cerebral palsy (CP). It improves participation, but not quality of life in children and adolescents with CP. It is more effective than usual care or other interventions in improving mobility, aerobic capacity, and balance in CP. It is no more effective than usual care or other interventions in improving muscle strength and spasticity.
[40]	PubMed, Web of Science, Scopus, OVID, Science Direct, PEDro, ERIC and ClinicalTrials.gov.	Twenty reports (16 RCTs); 12 to 64 participants per study.	Cerebral palsy, age ≤18 years.	Progressive resistance exercise.	Muscle strength was the primary outcome.	GMFM; MAS; FMS; PedsQL; 6MWT; MobQue; LifeHABITS; Harris Children’s SelfConcept Scale; PODCI; MTS; SPPC.	When PRE was compared with another therapy, there were no differences between groups. PRE is safe and increases muscle strength in young people with CP, which is maintained after training stops. The increase in muscle strength is unrelated to the PRE intensity or dose.
[33]	PubMed, SciELO, PEDro, ERIC, and Cochrane.	Five studies (four RCTs); 13 to 29 participants per study.	Cerebral palsy, 7–18 years.	Aerobic training.	Improve walk distance and VO2max.	6MWD or VO2max (maximum oxygen volume).	Positive effects on the cardiorespiratory fitness system in CP. Aerobic training statistically increased VO2 max in CP. Aerobic training increased the distance covered during the 6MWT.
[26]	PubMed, Web of Science, Scopus, Physiotherapy Evidence Database (PEDro), Google Scholar, and Proquest.	Eleven studies; 1 to 29 participants per study.	Cerebral palsy, 3–21 years.	Aquatic exercise.	Main outcomes: gross motor skills (walking speed or distance).	GMFM; COPM; Timed walk tests; PEDI; mobility and self-care domains.	The findings indicate that the effects of aquatic intervention on gross motor skills and walking speed are variable in children with CP who are ambulatory. The level of evidence is low for the majority of studies, especially for children classified according to GMFCS levels III–V, and further research is needed.
[32]	MEDLINE-PubMed, Cochrane, PEDro, CINAHL, and SPORTDiscus.	Twenty-seven studies; 873 participants.	Cerebral palsy, 3–22 years.	Strength training.	Strength, spasticity, gait, balance, energy expenditure, and motor function.	GMFM; daily step; Sitting time MAS; Dynamic Balance (FRT, PBS); WB-SI; CAPE; FEF; PedsQoLCP; PierrsHarris Self Concept Scale; MobQues28; COPM; STS; LSU; MAS; MMT; TUG; PEDI; FMS; FWT; 1MFWT; 6MWT; 10MWT;TST; NNcost.	Strength training programs appear to improve the strength of trained muscles, balance, gait, and motor function in CP, without increasing spasticity, but this type of intervention may demonstrate different effects in different neurological conditions.
[29]	PubMed and Embase.	Twenty-six studies; 3 to 40 participants per study.	Cerebral palsy, 6–18 years.	Virtual reality.	Balance, walking speed, and walking distance.	mABC-2; 1MFWT; 2MFWT; 6MWT; 10MWT; Quiet stance (force plate); Romberg test; CB&M TUDS; BOTMP; PBS; BBS; TUG; FAB; Dynamic stance (force plate); PRT; BOT 2; Msot; Reactive balance (force plate); Rhythmic weight shift test; FFRT; FSRT; TGGT.	VRT appears to be a promising intervention for rehabilitation in children with CP. Meta-analysis confirmed this positive effect. These results should be interpreted with caution due to differences in the interventions used, the lack of randomized controlled trials, and the relatively small groups.
[28]	PubMed, MEDLINE, Scopus, Web of Science, PEDro (Physioterapia Evidence Database), and SciELO.	Eleven RCTs; 16 to 40 participants per study.	Cerebral palsy, 5–16 years.	Nintendo Wii therapy.	Balance.	TGUGT; PBS; OLST.	NWT can be considered an effective treatment for improving functional and dynamic balance in children with CP, especially when combined with CPT in 30 min sessions, with interventions lasting longer than 3 weeks. Moderate-quality evidence.
[31]	CNKI, Wanfang Data, Web of Science, PubMed, EBSCOhost, Informit, Scopus, Science Direct, and ProQuest.	Eleven studies; 16 to 48 participants per study.	Cerebral palsy, ≤14 years.	Virtual reality.	Balance.	M-ABC; PBS; BBS; Nintendo Wii Fit Balance; Board Score; TUG; PBM.	VR games played a positive role in improving the balance of children with CP, but these results should be viewed with caution due to current methodological defects (measurement difference, heterogeneity of control groups, intervention combined with other treatments).
[25]	CINAHL, Embase, PubMed, Informações Psicológicas, Escopo, and Web of Science.	Sixteen studies; 1 to 27 participants per study.	Cerebral palsy, 7–18 years.	Movement and step-based dance practice.	Impact on walking/gait, balance/postural control, and range of motion.	PBS; PRT; 10MWT; GMFM; BCS; BBS; TUG; DGI; K-MBI; FGA; GAITRite system; TUDS; QUEST.	The review suggested a positive impact on the main outcomes but the findings should be interpreted with caution (limitations).
[30]	PubMed, PEDro, Web of Science, OTseeker, PsycINFO, and Cochrane.	Thirty-one studies; 1 to 32 participants per study.	Cerebral palsy, mean age 6–16.	Virtual reality.	Balance and motor skills.	GMFM; TUG; PBS; PEDI; Modified UE Functional Targeting Reach Test; TUDS; WeeFIM–The; QUEST; STST; PRT, RSA; AMPS; BBS; FMA; GAS; PSA.	Moderate evidence that VRT is a promising intervention to improve balance and motor skills in children and adolescents with CP. The technique is growing, so long-term follow-up and further research are required to determine its exact place in the management of cerebral palsy.
[27]	Google Acadêmico, PubMed and PEDRO	11 studies (6 RCTs). 5 to 17 participants per study	Cerebral palsy, 3–15 years	Aquatic interventions	Motor function, quality of life, fatigue, spasticity, balance, enjoyment, behaviour, activities of daily living, strength, endurance and aerobic capacity.	GMFM	The review suggest that aquatic therapy has the potential to achieve better motor functions in CP patient with GMFCS levels between I–III when compared with conventional land-based therapy or to no intervention.

1MFWT: 1-Minute Fast Walking Test, 2MFWT: 2-Minute Fast Walking Test, 6MWT: 6-Minute Walking Test, 6MWD: 6 Min Walk Distance, 10MWT: 10-Metre Walking Test, AMPS: Assessment of Motor and Process Skills, ASK: Activity Scale for Kids, BBS: Berg Balance Scale, BOT 2: Bruininks–Oseretsky test of Motor Proficiency 2nd Edition, BOTMP: Bruininks–Oseretsky test of Motor Proficiency, BCS: Body Cathexis Scale, COPM: Canadian Occupational Performance Measure, CAPE: Children’s Assessment of Participation and Enjoyment, CP: QoL Children–Cerebral Palsy–Quality of Life Children, CB&M: Community Balance and Mobility Scale, DGI: Dynamic Gait Index, FEF: Forced Expiratory Flow, FMS: Functional Mobility Scale, FMA: Fugl–Meyer Assessment, FSRT: Functional Sideways Reach Test, FFRT: Functional Forward Reach Test, FRT: Functional Reach Test, FGA: Functional Gait Assessment, FIM: Functional Independence Measure, FAB: Fullerton Advanced Balance Scale, GAS: Goal Attainment Scaling, GMFM: Gross Motor Function Measure, GMFCS: Gross Motor Function Classification System, ICF: International Classification of Functioning, K-MBI: Modied Barthel Index, WeeFIM: Functional Independence Measure for Children, LSU: Lateral Step Up, LifeHABITS (Life-H): Life Habits Questionnaire, MAS: Modified Ashworth Scale, MobQues28: Mobility Questionnaire, MMT: Manual Muscle Test, MTS: Modified Tardieu Scale, mABC-2: Movement Assessment Battery for Children-2, mSOT: Modified Sensory Organization Test, NNcost: Net Nondimensional Oxygen Cost, OLST: One Leg Stance Test, PBS: Pediatric Balance Scale, PEDI: Pediatric Evaluation of Disability Inventory, PRT: Paediatric Reach Test; PSA: Posture Scale Analyzer, PBM: Pediatric Balance Measurement, PedsQL: Pediatric Quality of Life Scale, PODCI: Pediatric Outcomes, PEDI-CAT: Pediatric Evaluation of Disability Inventory Computer Adaptive Test, PEDALS: Pediatric Evaluation of Disability Inventory, PEM-CY: Participation and Environment Measure—Children and Youth, QUEST: Quality of Upper Extremity Skills Test, RSA: Running Speed and Agility, RCTs: Randomized Controlled Trial, STS: Sit-To-Stand, STST: Sit-To-Stand Test, SF-36: Short Form Survey Instrument, SPPC: Self-Perception Profile for Children, TUDS: Timed Up and Down Stairs, TST: Timed Stairs Test, TUG: Time Up and Go test, TGUGT: Timed Get Up and Go Test, WB-SI: Weight Bearing–Simmetry Index, WHOQOF-BREF: World Health Organization Quality of Life BREF.

**Table 2 healthcare-12-02039-t002:** AMSTAR2 results.

Reference Number	Design	1	2	3	4	5	6	7	8	9	10	11	12	13	14	15	16	Rating
[24]	MA	Y	Y	Y	PY	N	Y	Y	Y	PY	N	Y	N	N	Y	N	Y	CL
[42]	MA	Y	Y	Y	PY	Y	Y	N	Y	Y	N	N	N	N	N	N	Y	CL
[34]	SR	Y	Y	Y	Y	Y	Y	Y	PY	Y	N	NM	NM	Y	Y	NM	Y	H
[35]	MA	Y	PY	N	PY	Y	Y	Y	PY	N	N	Y	N	N	Y	N	Y	CL
[36]	MA	Y	Y	Y	PY	Y	Y	Y	PY	Y	N	Y	Y	Y	Y	Y	Y	H
[41]	MA	Y	Y	Y	PY	Y	Y	N	PY	Y	N	N	N	N	N	N	Y	CL
[39]	MA	Y	PY	Y	PY	Y	Y	Y	PY	Y	N	N	Y	N	N	N	Y	CL
[38]	SR	Y	Y	Y	Y	Y	Y	Y	Y	Y	N	NM	NM	Y	Y	NM	Y	H
[37]	MA	Y	Y	Y	PY	Y	Y	Y	N	Y	N	Y	Y	Y	Y	Y	N	M
[40]	MA	Y	PY	N	PY	Y	Y	Y	PY	Y	N	N	N	N	N	N	Y	CL
[33]	MA	Y	Y	Y	PY	Y	Y	N	PY	Y	N	Y	Y	N	N	N	Y	CL
[26]	SR	Y	PY	Y	PY	Y	Y	Y	PY	Y	N	NM	NM	N	N	NM	Y	L
[32]	MA	Y	Y	Y	Y	Y	Y	Y	Y	N	N	Y	Y	Y	Y	N	Y	CL
[29]	MA	Y	PY	Y	PY	Y	Y	Y	Y	PY	N	Y	N	Y	Y	N	Y	L
[28]	MA	Y	Y	Y	Y	Y	Y	Y	PY	Y	Y	Y	N	Y	Y	Y	N	M
[31]	MA	Y	Y	Y	PY	Y	Y	Y	Y	Y	N	Y	N	Y	N	Y	Y	M
[25]	MA	Y	Y	N	PY	Y	Y	Y	PY	Y	N	N	N	N	N	N	Y	CL
[30]	SR	Y	Y	N	PY	Y	Y	N	Y	Y	Y	NM	NM	Y	Y	NM	Y	L
[27]	SR	Y	PY	N	PY	N	N	Y	Y	N	N	NM	NM	N	N	NM	Y	CL

## Data Availability

The authors confirm that the data supporting the findings of this study are available within the article and its Appendix A.

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
