# Peer review of "The Methodological Quality of Studies on Physical Exercise in Adolescents with Cerebral Palsy: A Scoping Review of Systematic Reviews and Meta-Analyses"

_healthcare, 2024, doi:10.3390/healthcare12202039_

Round 1
Reviewer 1 Report
Comments and Suggestions for Authors
I commend the authors for conducting and writing this research. The topic is important and highly relevant. Contributing to a deeper understanding of studies investigating the practice of physical exercise in cerebral palsy, the leading cause of physical disability in childhood, is indeed valuable.
However, despite the relevance and interest of the topic, several weaknesses lead me to recommend discontinuing the review process. In my view, these issues are critical and negatively impact the quality of the study and the interpretation of the results. Below, I list the three main points that led me to this recommendation, followed by other secondary concerns.
(1) The PICOs question is clearly stated in lines 96 and 97: "(i) Patients: systematic reviews or meta-analyses that included articles where at least 50% of the sample comprised adolescents (12-18 years)." However, when examining the selected studies, we find extremely broad age ranges (e.g., 1–21 years and 0–22 years). In some studies, the majority of the population in the target groups are children and infants. Therefore, extrapolating the effects and results of physical exercise protocols across such a wide age range is imprudent and inconsistent.
(2) Regarding the PICOs, the intervention specified was physical exercise. However, a substantial portion of the studies investigates physiotherapeutic interventions or rehabilitation conducted by physiotherapists. This item clearly needs to be revised or reformulated.
(3) The databases listed in the methods section are PubMed, Cochrane, and Web of Science. However, in Table 1, the database column includes several other databases and appears to reference gray literature. The information about the databases used must be clarified, along with the search strategies.
-
It is unclear whether the evaluators had formal training in AMSTAR-2 and GRADE; this information should be included in the text.
-
Section 3.2.2 would benefit from rewriting. It is difficult to follow, and the studies are not cited, making it hard to understand which studies the authors are referring to.
-
Adding the reference number next to the author's name in Table 1 would improve the readability of the text.
The English is generally good, but the text would benefit from further revision to enhance readability.
Author Response
(1) The PICOs question is clearly stated in lines 96 and 97: "(i) Patients: systematic reviews or meta-analyses that included articles where at least 50% of the sample comprised adolescents (12-18 years)." However, when examining the selected studies, we find extremely broad age ranges (e.g., 1–21 years and 0–22 years). In some studies, the majority of the population in the target groups are children and infants. Therefore, extrapolating the effects and results of physical exercise protocols across such a wide age range is imprudent and inconsistent.
Response: Dear reviewer, we would like to thank you for the opportunity to rewrite this topic, making it more transparent and easier to understand. As rewritten in the text, we selected systematic reviews and meta-analyses for our review, in which at least 50% of the included articles had adolescents (10 to 19 years old) with CP in their sample.
Initially, we planned to research ages 12 to 18, but we adjusted the age range to follow the range considered adolescents by the World Health Organization. We appreciate your alertness and thank you for bringing this to our attention. We are sorry for only doing it now.
(2) Regarding the PICOs, the intervention specified was physical exercise. However, a substantial portion of the studies investigates physiotherapeutic interventions or rehabilitation conducted by physiotherapists. This item clearly needs to be revised or reformulated.
Response: Dear reviewer, all systematic reviews and meta-analyses that included articles comparing exercise before and after the intervention or comparing exercise with any other intervention were considered, as long as the target population received an intervention with some physical exercise program. We explain these details in item 2.3.3 – comparison.
(3) The databases listed in the methods section are PubMed, Cochrane, and Web of Science. However, in Table 1, the database column includes several other databases and appears to reference gray literature. The information about the databases used must be clarified, along with the search strategies.
Response: Dear reviewer, we apologize for not being clear enough. Table 1 shows the articles found in the databases mentioned above. Such articles are systematic reviews or meta-analyses. This table includes details of the 19 reviews/meta-analyses and the databases where they searched for the articles they included in their own reviews. Some even report that they included gray literature. The fact that systematic reviews and meta-analyses include gray literature is one of the reasons cited by the authors of the AMSTAR 2 instrument to justify its use (Shea et al., 2017).
(4) It is unclear whether the evaluators had formal training in AMSTAR-2 and GRADE; this information should be included in the text.
Response: Dear reviewer, the authors of the Amstar 2 instrument provide free use of the instrument, a page with frequently asked questions and answers, and an address for clarifying doubts. There is no formal training to use the instrument at the moment. However, the instrument's official page informs that a training video will soon be made available (https://amstar.ca/index.php).
As for GRADE, the authors did not undergo formal training. They participated in the workshop offered by the Diretrizes platform, Development of clinical-care guidelines for SUS Brazil. As it is an open course, without attendance records or proof of participation, it was not added to the text.
(5) Section 3.2.2 would benefit from rewriting. It is difficult to follow, and the studies are not cited, making it hard to understand which studies the authors are referring to.
Response: Dear reviewer, we have added the information (section 3.2.1 and 3.2.2)
(6) Adding the reference number next to the author's name in Table 1 would improve the readability of the text.
Response: Dear reviewer, we have added the information
(7) Comments on the Quality of English Language
The English is generally good, but the text would benefit from further revision to enhance readability.
Response: Dear reviewer, all the authors have rechecked the text and improved it. Thank you for your collaboration and for helping us improve our work.

Reviewer 2 Report
Comments and Suggestions for Authors
Thank you for the opportunity to review the article: “The methodological quality of studies on physical exercise in adolescents with cerebral palsy: a scoping review of systematic reviews and meta-analyses”.
This manuscript is a scoping review that analyses the methodological quality of systematic reviews and meta-analyses on the effects of physical exercise in adolescents with cerebral palsy. The main objective is to assess the methodological limitations of the research and to identify priorities for future studies.
I will now proceed to analyse its content and provide my comments and observations to contribute to its improvement and strengthening.
- Introduction: In a scientific manuscript, it is more common to use the term ‘Introduction’ rather than ‘Background’ when it comes to the main section of the text. The term ‘Background’ is often used in abstracts or informal discussions, but in the body of the article, ‘Introduction’ is the standard term for the opening section.
It would be more appropriate to replace ‘Background’ with ‘Introduction’.
- Although it is mentioned that the study focuses on adolescents with cerebral palsy, it would be useful to explain why this particular group is the focus of the research. Is there anything unique or particularly relevant about adolescence that makes this population require a special focus in terms of physical exercise?
-Material and Methods: although the search terms are well defined, you could be more explicit on how these terms were combined or if additional hand searching of the reference lists of key studies was done. You could also mention if there were any language or formatting limitations in the search.
- In most cases, when quoting a record number in a record such as PROSPERO, it is sufficient to provide the record number without the need to specify ‘ID’. This is because the context already makes it clear that it is an identifier for a database entry.
- Physical activity and exercise are distinct concepts and, in a scientific paper, it is important that the terms used in the search are aligned with the concepts presented in the introduction and the rest of the article. If the introduction does not clearly explain the distinction between these terms or does not adequately mention physical activity, confusion may arise.
Physical activity is any bodily movement produced by skeletal muscles that results in energy expenditure (including walking, manual labour, etc.), while exercise is a subcategory of physical activity that involves structured, repetitive movements with the aim of improving or maintaining physical fitness.
If the authors use both terms in the search, but haven´t justified this decision in the introduction or methodology, the reason should be made clear.
- I don't understand why there is a checklist in the annex.
- Results: The legend in table 1 is missing. It is advisable to use abbreviations in the table and then include the legend to explain the meaning of the acronyms. For example:
Nintendo Wii Therapy → NWT
Conventional Physical Therapy → CPT
Also in table 1, the term ‘n varying’ is not understood.
- Add a column indicating only othe type of exercise used in each study (e.g., ‘Aerobic exercise’, ‘Resistance therapy’, ‘Virtual reality exercise’) and another column could contain the variables studied and the systems or scales used to measure the results (e.g., balance, muscle strength, gait speed, using Berg's scale to measure balance).
-Discussion: I suggest that including the limitations within the discussion section could be a valid option, as it allows the reader to have a more integrated view of the strengths and weaknesses of the study. It becomes clearer how they affect the interpretation of the results and recommendations for future research.
- In acknowledgements, the mention of institutions is appropriate, but to be more specific, it would be useful to explain briefly why these institutions are being thanked.
- Authors’ contributions: it is unusual to see the mention of ‘Authors 1, 2, 3, etc.’ instead of using the initials of the authors' names, as is the recommended practice in many scientific style guides (e.g. Vancouver or APA). Normally, the standard format would include the initials to provide clarity on who contributed to which part of the study; this structure provides greater transparency and makes it easier to identify the individual contributions of each author.
Author Response
(1) Introduction: In a scientific manuscript, it is more common to use the term ‘Introduction’ rather than ‘Background’ when it comes to the main section of the text. The term ‘Background’ is often used in abstracts or informal discussions, but in the body of the article, ‘Introduction’ is the standard term for the opening section.
It would be more appropriate to replace ‘Background’ with ‘Introduction’.
Response: Dear reviewer, thank you for your observation. We made the amendment.
(2) Although it is mentioned that the study focuses on adolescents with cerebral palsy, it would be useful to explain why this particular group is the focus of the research. Is there anything unique or particularly relevant about adolescence that makes this population require a special focus in terms of physical exercise?
Response: Dear reviewer, thank you for your observation. We briefly add to the introduction our particular interest in this study's target population: "During the transition to adulthood, young people with CP experience decreasing physical activity levels [19]."
(3) Material and Methods: although the search terms are well defined, you could be more explicit on how these terms were combined or if additional hand searching of the reference lists of key studies was done. You could also mention if there were any language or formatting limitations in the search.
Response: Dear reviewer, thanks for the opportunity to clarify this point. Initially, we broadly searched using synonyms, variants, and related terms. However, the search yielded an enormous scope beyond our research object. In this way, we chose to restrict the terms to reach the real object of the research and make a more significant contribution to academic production. On line 87 are language and time limitations.
(4) In most cases, when quoting a record number in a record such as PROSPERO, it is sufficient to provide the record number without the need to specify ‘ID’. This is because the context already makes it clear that it is an identifier for a database entry.
Response: Dear reviewer, thank you for your observation. We made the amendment.
(5) Physical activity and exercise are distinct concepts and, in a scientific paper, it is important that the terms used in the search are aligned with the concepts presented in the introduction and the rest of the article. If the introduction does not clearly explain the distinction between these terms or does not adequately mention physical activity, confusion may arise.
Response: Dear reviewer, thank you for your observation. We carefully reviewed the text and made adjustments. We are sorry for only doing it now, and thank you for the alert.
(6) Physical activity is any bodily movement produced by skeletal muscles that results in energy expenditure (including walking, manual labour, etc.), while exercise is a subcategory of physical activity that involves structured, repetitive movements with the aim of improving or maintaining physical fitness.
If the authors use both terms in the search, but haven´t justified this decision in the introduction or methodology, the reason should be made clear.
Response: Dear reviewer, we appreciate the opportunity to clarify this issue. Our study considered all systematic reviews and meta-analyses that included articles comparing exercise before and after intervention or comparing exercise with physical activity. For this reason, the term physical activity was included in the search. We explain these details in item 2.3.3 – comparison.
(7) I don't understand why there is a checklist in the annex.
Response: Dear reviewer, as the journal requests non-publishable supplementary material with the review, we sent the checklist of the main instrument.
(8) Results: The legend in table 1 is missing. It is advisable to use abbreviations in the table and then include the legend to explain the meaning of the acronyms. For example:
Nintendo Wii Therapy → NWT
Conventional Physical Therapy → CPT
Also in table 1, the term ‘n varying’ is not understood.
Response: Dear reviewer, thank you for your observation. We made the amendment.
(9) Add a column indicating only other type of exercise used in each study (e.g., ‘Aerobic exercise’, ‘Resistance therapy’, ‘Virtual reality exercise’) and another column could contain the variables studied and the systems or scales used to measure the results (e.g., balance, muscle strength, gait speed, using Berg's scale to measure balance).
Response: Dear reviewer, thank you for your observation. We made the amendment.
(10) Discussion: I suggest that including the limitations within the discussion section could be a valid option, as it allows the reader to have a more integrated view of the strengths and weaknesses of the study. It becomes clearer how they affect the interpretation of the results and recommendations for future research.
Response: Dear reviewer, thank you for your observation. We made the amendment.
(11) In acknowledgements, the mention of institutions is appropriate, but to be more specific, it would be useful to explain briefly why these institutions are being thanked.
Response: Dear reviewer, thank you for your observation. Considering there is no particular reason, we removed the topic. (12) Authors’ contributions: it is unusual to see the mention of ‘Authors 1, 2, 3, etc.’ instead of using the initials of the authors' names, as is the recommended practice in many scientific style guides (e.g. Vancouver or APA). Normally, the standard format would include the initials to provide clarity on who contributed to which part of the study; this structure provides greater transparency and makes it easier to identify the individual contributions of each author.
Response: Dear reviewer, thank you for your observation. We made the amendment.

Reviewer 3 Report
Comments and Suggestions for Authors
Thank you for the opportunity to review this paper. This is a timely issue to explore. The aim of this study was to evaluate the methodological quality of systematic reviews and meta-analyses on the effects of physical activity in adolescents with CP to elucidate the methodological limitations of the research, seek continuous improvement of the review processes on the topic, and identify priorities to be observed in future research. It is clear paper, there is, however, issues that must be resolved before the study can be accepted for publication. Several comments and suggestions for the authors.
- Authors should also remember to exclude so-called grey literature from their research.
- Please correct the references, e.g. not [14,15,16,17] but [14-17], not [32,33,34,35,36,37,38,39,40,41,42] but [32-42] etc.
- Research articles usually do not use the word „we”, „our” and regularly use passive verbs.
Author Response
(1) Authors should also remember to exclude so-called grey literature from their research.
Response:
Dear reviewer, we apologize for not being clear enough. Table 1 shows the articles found in the databases mentioned above. Such articles are systematic reviews or meta-analyses. This table includes details of the 19 reviews/meta-analyses and the databases where they searched for the articles they included in their own reviews.
(2) Please correct the references, e.g. not [14,15,16,17] but [14-17], not [32,33,34,35,36,37,38,39,40,41,42] but [32-42] etc.
Response:
Dear reviewer, thank you for your observation. We made the amendment. We are sorry for only doing it now, and thank you for the alert.
(3) Research articles usually do not use the word „we”, „our” and regularly use passive verbs.
Response:
Dear reviewer, thank you for your observation. We made the amendment.